# Effect of the Propionylation Method on the Deformability under Thermal Pressure of Block-Shaped Wood

**DOI:** 10.3390/molecules26123539

**Published:** 2021-06-10

**Authors:** Mitsuru Abe, Masako Seki, Tsunehisa Miki, Masakazu Nishida

**Affiliations:** Multi-Materials Research Institute, National Institute of Advanced Industrial Science and Technology (AIST), 2266-98 Shimoshidami, Moriyamaku, Nagoya 463-8560, Japan; m-seki@aist.go.jp (M.S.); tsune-miki@aist.go.jp (T.M.); m-nishida@aist.go.jp (M.N.)

**Keywords:** wood, thermoplasticity, esterification, propionylation, mechanism, deformability, hot-press

## Abstract

Converting wood waste into thermoplastic materials is an attractive means of increasing its utilization because complex three-dimensional molded products can easily be obtained by press molding wood with thermoplasticity. Chemical modification, especially esterification, is a promising method for imparting thermoplasticity to wood. In this study, we produced multiple propionylated wood specimens using several propionylation methods and elucidated the factors affecting the deformability of the wood. Regardless of the method, all of the propionylated wood samples showed deformability in the tangential direction. However, in the longitudinal direction, not only the degree of propionylation but also the propionylation method had a significant influence on the deformability. The flow in the tangential direction occurred under a relatively low pressure, whereas the flow in the longitudinal direction occurred under higher pressure. The chemical composition and motility of each sample were evaluated using solid-state NMR measurements. With some propionylation methods, decomposition of the cellulose main chain occurred during the reaction, which had a dominant effect on the deformability of the wood in the longitudinal direction. These results indicate that the deformability of wood can be controlled by the appropriate selection of a propionylation method and its treatment conditions.

## 1. Introduction

Cutting, scraping, bending, and compression techniques are typically used to obtain two-dimensional and/or three-dimensional materials from wood. Plastic formation from wood has recently received significant attention as a new method for producing three-dimensional materials. For example, wood–plastic composites (WPCs), which are made by kneading wood flour and plastic, are being actively investigated as raw materials for plastic formation [1,2,3]. However, to produce such a WPC, wood is used in powdered form. This powdering pretreatment consumes both time and energy. Moreover, during this pretreatment, a certain amount of the mechanically beneficial wood fiber structure is lost.

To overcome these shortcomings, a new method for plastic formation from wood has been developed. This method does not require the use of powdered wood or the kneading of plastic and wood flour. In this method, wood with a certain amount of water is induced to flow through heating and compression [4,5]. In traditional wood compression, wood cell lumens disappear, but the positional relationship between the wood cells does not change. In the new method, wood flow occurs through the sliding of each cell wall under certain conditions. Using this new flow-forming method, the one-shot production of a complex molded body is possible. Recently, the deformability of wood, which is a measure of how easily the wood flows under low temperature and/or pressure, has been found to be improved by impregnating the wood with a monomer such as a phenol or melamine resin [6,7]. Furthermore, by curing the resin monomer through heat treatment, water resistance could be successfully imparted to the flow-formed product. In addition, the dimensional stability of the product was also improved. However, because these resins are thermoset and cannot be re-formed, their recyclability remains a challenging issue. Therefore, a monomer of methyl methacrylate (MMA), which is a thermoplastic resin, was used as the binder. As a result, thermoplastic flow-formed products that could be re-formed were developed [8,9,10]. However, water cannot be used as a solvent in the impregnation solution because MMA is highly hydrophobic; therefore, it is necessary to use an organic solvent. An alternative method is chemical modification of the wood surface. The affinity between the wood and MMA can be improved by substitution of the hydroxyl groups in the wood with acetyl groups. Although these pretreatment methods have shown some positive results, the complexity of the treatment process requires further improvement. Additionally, these methods require large amounts of petroleum-derived resins. Therefore, the development of additional novel methods is required.

The synthesis of wood thermoplastics through derivatization is considered a promising solution. Wood flour has been reported to exhibit thermal fusibility via esterification treatment [11,12,13,14]. Shiraishi et al. succeeded in obtaining a film from an esterified wood powder [12]. Recently, several research groups have developed thermoplastics through the esterification of wood. Xie et al. reported an efficient esterification process for lignocellulosic materials in ionic liquids [15]. They reported that acetylated wood showed a clear glass transition temperature at 131.7 °C. Su et al. successfully obtained injection-molded wood-based plastics through a simple reaction of wood flour with phthalic anhydride in 1-methylimidazole at 15 °C for 2 h [16]. Chen et al. developed a translucent film through the esterification of wood powder in a mixture of an ionic liquid and dimethyl sulfoxide [17]. Hassan et al. succeeded in obtaining bagasse fibers with thermal fusibility through esterification treatment with carboxylic anhydride and acetone [18,19]. Furthermore, they stated that the use of fibrous biomass instead of wood flour affords the possibility of obtaining a novel material that utilizes the strength of the fibers in plant biomass. They analyzed the surface condition of hot-melted bagasse; however, they did not report the three-dimensional flow formed using esterified bagasse. In addition, for block-shaped esterified wood, several thermophysical properties such as dimensional stability and dynamic viscoelasticity have been reported [20]; however, to the best of our knowledge, there have been no reports of fluid molding until very recently.

Recently, we succeeded in rendering thermoplasticity to block-shaped wood by introducing ester groups using trifluoroacetic anhydride (TFAA) and carboxylic acids [21]. The propionylated wood flowed more easily during hot pressing than acetylated wood. Additionally, a three-dimensional cup-shaped molded product was obtained by heat press molding of the propionylated wood. The obtained product had water-repellent characteristics, and it could be molded multiple times owing to its thermoplasticity. However, many aspects of the relationship between wood esterification treatment and the deformability of wood under hot pressing remain unclear.

In this study, block-shaped wood was propionylated using several methods (Table 1) to investigate the expression mechanism of thermoplasticity and other characteristics. The chemical composition and deformability during hot pressing of the obtained woods were analyzed to clarify the correlation between the type of propionylation method and the wood deformability, including the degree and direction.

## 2. Results and Discussion

### 2.1. Wood Propionylation

Photographs of the degreased and propionylated wood samples are shown in Figure 1. Table 2 lists the increase in the weight and length of the propionylated wood pieces relative to the degreased wood. For the length of the sides, the central part of the sample was measured. There was almost no change after the propionic anhydride (PA) treatment. The **TFAA****-W** sample was distorted and significantly increased in weight. In addition, swelling in the T-direction and contraction in the L-direction were confirmed. In **DMAP****-W_1,2**, distortion similar to that of the **TFAA****-W** specimen did not occur, but the **DMAP****-W_1,2** samples expanded slightly in the T-direction. On the other hand, in **SA****-W_1-4**, the wood pieces shrank and became brown in color as the treatment time increased. In addition, fine cracks were observed on the wood surface. The conditions of the propionylated woods were found to differ depending on the propionylation method.

### 2.2. FT-IR Measurement

Figure 2 shows the Fourier-transform infrared (FT-IR) spectra of the degreased and propionylated wood samples in the dry state. The IR peaks of the wood samples were assigned by referencing the report by Mohebby [22]. The peak intensity of the hydroxyl groups (3000–3600 cm^−1^) decreased significantly after the propionylation treatment. On the other hand, IR peaks attributable to ester groups appeared at 1740 cm^−1^ after propionylation. These changes indicate that the hydroxyl groups were replaced with propionyl groups. To estimate the reactivity of the propionylation, the relative peak intensities of the FT-IR spectra were analyzed. The strength of the band derived from the C–O stretching vibration appearing at 1060 cm^−1^ (Figure 3a) was used as a reference because its intensity did not change notably before and after propionylation. The decrease in the OH-derived peak was calculated as the ratio of the peak intensity derived from the OH stretching vibrations (Figure 3b) to the peak intensity of the C–O stretching vibrations (OH/C–O intensity ratio). The increase in the peaks derived from the ester groups was calculated as the ratio of the peak intensity derived from the C=O stretching vibration at 1740 cm^−1^ (Figure 3c) to the peak intensity derived from the C–O stretching vibrations (C=O/C–O intensity ratio). The smaller the OH/C–O intensity ratio and larger the C=O/C–O intensity ratio, the greater the number of hydroxyl groups replaced with propionyl groups. The calculated OH/C–O intensity ratio of the degreased wood was 0.27, and the C=O/C–O intensity ratio was 0.13, whereas the corresponding values for **TFAA****-W** were 0.07 and 0.70, respectively.

Figure 4 shows the relationship between the OH/C–O intensity ratio and the C=O/C–O intensity ratio. There was almost no effect due to the difference in propionylation method, and there was a close relationship between the decrease in OH/C–O intensity ratio and the increase in C=O/C–O intensity ratio. These results show that the progress of the wood propionylation reaction can be estimated based on both the OH/C–O intensity ratio and the C=O/C–O intensity ratio. In the following sections, the discussion will proceed mainly using the C=O/C–O intensity ratio, because this ratio is thought to provide a direct indication of the content of propionyl groups.

Figure 5 shows the relationship between the C=O/C–O intensity ratio and the density of the propionylated wood specimens. The density of the wood tended to increase as the propionylation progressed. The difference in the propionylation method did not affect the density of the obtained wood.

### 2.3. Hot Compression Test

Figure 6 shows actual observations of the pressed samples. The sample pieces before pressing were 5 × 5 × 5 mm cubes, and their initial size is indicated by the dotted line in the image of the **DGR-W** sample. In the **DGR-W** and **PA****-W** samples, a slight spreading was observed only in the T-direction. In contrast, all of the other propionylated cypress samples were stretched in the T-direction. In particular, **TFAA****-W** and **SA****-W_4** expanded not only in the T-direction but also in the L-direction.

Next, the relationship between the degree of propionylation of the wood and the deformability during hot pressing was evaluated. The degree of propionylation was evaluated using the C=O/C–O intensity ratio, and the deformability was evaluated as the area per unit weight of the pressed specimens (Figure 7). The **TFAA****-W** and **SA****-W_4** specimens were significantly stretched by pressing, and the area per unit weight of the pressed specimens was ~1.9 mm^2^/mg. Specimens **SA****-W_1**–**3** were also slightly stretched; however, their stretching ratios were ~1.2–1.4 mm^2^/mg. In the case of **DMAP****-W_1** and **2**, the areas per unit weight were almost the same as that of **DGR****-W** and **PA****-W**, ~1 mm^2^/mg. Although **DMAP****-W_1** and **2** had similar C=O/C–O intensity ratios to **TFAA****-W** and **SA****-W_4**, their deformability was very low. Therefore, the deformability of wood was found to be strongly influenced not only by the degree of propionylation but also by the propionylation method.

Figure 8 shows the compressive stress–strain curves of the degreased and propionylated woods during hot pressing. No yield stress was observed in the degreased specimen. On the other hand, yield stresses were observed for all of the propionylated cypresses (Figure 8, arrows). Two yield stresses appeared for specimens **TFAA****-W** and **SA****-W_4**, but only one yield stress was observed for the other propionylated samples, including **DMAP****-W_1** and **DMAP****-W_2**, which had almost the same degree of propionylation as **TFAA****-W** and **SA****-W_4**. Both the propionylation method and degree of propionylation may influence the appearance of multiple yield stresses. In the following paragraphs, the yield stress observed in all propionylated wood specimens is defined as the first yield stress, and that observed in only **TFAA****-W** and **SA****-W_4** under relatively high-pressure conditions is defined as the second yield stress.

From the results shown in Figure 6, the first yield stress may be caused by stretching in the T-direction, and the second yield stress may be caused by stretching in the L-direction. To confirm this hypothesis, **TFAA****-W** specimens pressed at different pressures were prepared. The **TFAA****-W** test pieces pressed with a maximum load of ~80 MPa or ~200 MPa were designated as **TFAA****-W_80MPa** and **TFAA****-W_200MPa**, respectively. **TFAA****-W_200MPa** was extended in both the T- and L-directions. On the other hand, **TFAA****-W_80MPa** was only extended in the T-direction and hardly extended in the L-direction (Figure 9). Further, microscopic observations were conducted on the two **TFAA****-W** pressed pieces. In **TFAA****-W_80MPa**, the shape of the fibers lined up in the L-direction was maintained after compression. On the other hand, many fibers were torn in specimen **TFAA****-W_200MPa**. Therefore, the first and second yield stresses are thought to be derived from the flow in the T- and L-directions, respectively.

Next, the relationship between the degree of propionylation and the value of the yield stress was evaluated. The value of the first yield stress tended to decrease as the C=O/C–O intensity ratio increased (Figure 10). The same tendency was observed even with different propionylation methods. This indicates that stretching in the T-direction occurred under lower pressure as the propionylation progressed, regardless of the propionylation method. Regarding the second yield stress, specimen **SA-W_4**, which had a higher degree of propionylation, showed a lower value of the yield stress, indicating that the pressure required for stretching in the L-direction was lower than that for **TFAA-W**. However, it is difficult to draw a clear conclusion on the correlation between the degree of propionylation and the value of the yield stress because only a small number of samples exhibited the second yield stress.

### 2.4. Solid-State NMR Measurements

In this section, five propionylated woods with relatively high degrees of propionylation are evaluated using the solid-state nuclear magnetic resonance (NMR) method, with degreased wood as the reference. ^1^H magic-angle spinning (MAS) NMR can directly monitor not only the propionylation process but also the affinity of the propionylated wood for water molecules. Figure 11 shows the ^1^H MAS NMR spectra of the propionylated wood specimens in the humid state after one week of storage at 20 °C and a relative humidity of 60%. The ^1^H MAS NMR spectra of sulfuric acid (SA)-propionylated wood specimens under various reaction conditions are shown in Appendix A. Before propionylation, the degreased wood could hold a relatively large amount of water molecules, which appear at around 5 ppm (Figure 11, **DGR-W**). As the propionylation progressed, C_2_H_5_ protons appear as a broad signal at around 1 ppm. The reaction progress of the 4-dimethylaminopyridine (DMAP) treatment was found to be slower than that of SA treatment at the same treatment temperature and time (see Figure 11, **SA-W_4** and **DMAP-W_1**). Owing to the increase in hydrophobicity, both the TFAA- and DMAP-propionylated wood specimens contained only a small amount of water molecules, even in the humid state. However, the SA-propionylated wood contained a relatively large amount of water molecules at lower propionylation rates (Appendix A, **SA-W_1**–**3**), and the signal intensity of the water molecules rapidly decreased with the increasing propionylation rate (**SA-W_4**). Therefore, the DMAP-propionylated wood was hydrophobized in the early stages of propionylation, while the SA-propionylated wood was hydrophobized after the propionylation progressed.

Figure 12 and Appendix A show the ^13^C cross-polarization (CP)-MAS NMR spectra of degreased and propionylated wood specimens. Signals corresponding to biomass constituents in the degreased wood were assigned with reference to our previous report [23]. The signals of the introduced propionyl group appeared at 173 ppm (C=O), 28 ppm (CH_2_), and 9 ppm (CH_3_). The signal intensity of the introduced propionyl group was approximately consistent with the trend in the ratio of the IR spectra. For the constituent polymers in wood, the intensity of the aromatic ring signals of lignin decreased with the progression of propionylation in all samples. The rate of aromatic signal reduction was particularly high in the **TFAA-W** specimen. On the other hand, different tendencies were observed in the carbohydrate signal depending on how the sample was treated. In **TFAA-W** and **SA-W_4**, the cellulose C4 signal disappeared, and the cellulose C2, C3, and C5 signals became singlets. In addition, the cellulose C1 shifted to a higher magnetic field. These changes in the cellulose signals indicate that propionylation with both TFAA and SA treatments resulted in decomposition of the cellulose main chain. On the other hand, in the DMAP-propionylated wood, the cellulose C4 remained, and the cellulose C2, C3, and C5 signals did not become singlets. This indicates that almost no side reactions occurred in the cellulose main chain. According to the ^13^C CP-MAS NMR spectral changes with varying reaction times and temperatures (Appendix A), the lignin portion of the wood increased with increasing propionyl rate in the initial stage (**SA-W_1**–**3**). The increased lignin content, however, decreased with the disappearance of the cellulose C4 signal (**SA-W_4**). Considering the NMR results together with the results of the pressing experiments mentioned above suggests that the flow in the L-direction occurs only when the cellulose main chain is decomposed to some extent during the propionylation treatment.

The ^13^C pulse saturation transfer (PST)-MAS NMR method can provide information on molecular mobility related to the nuclear Overhauser effect (NOE). The ^13^C PST-MAS method enhances the signals of flexible components with high mobility near hydrogen atoms, such as alkyl or alkoxy side chains [24]. Figure 13 shows the ^13^C PST-MAS NMR spectra with the assignments of each substituent in the propionylated wood specimens. The ^13^C PST-MAS NMR spectra of SA-propionylated wood specimens under various reaction conditions are shown in Appendix A. Because the introduced propionyl group had higher molecular mobility than the carbohydrate chains, both the C_2_H_5_ and C=O groups showed higher signal intensities in the ^13^C PST-MAS NMR spectra. Compared with the ^13^C CP-MAS NMR spectra, propionylation reduced the signal intensities of the carbohydrate chains. Regarding the effect of the propionylation method on the ^13^C PST-MAS NMR spectra, TFAA- and DMAP-propionylated wood specimens had higher PST-MAS signal intensities for CH_2_ and CH_3_ groups than the SA-propionylated wood specimen based on the corresponding CP-MAS signal intensities. Furthermore, the ratio of the PST-MAS signal intensity to CP-MAS signal intensity was largest in specimen **SA-W_1**, which had the lowest propionyl content (Appendix A). Therefore, the molecular mobility of the C_2_H_5_ group was suppressed in the SA-propionylated wood compared with that in the TFAA- and DMAP-propionylated wood specimens. 

To examine the interactions between constituent polymers in the propionylated wood specimens, the *T*_1_H values in the dry state were measured, as shown in Figure 14. The treatment of wood with TFAA and DMAP increased the *T*_1_H values of the carbohydrates, while treatment with SA increased them slightly (Figure 14a). The different trend in the *T*_1_H increase due to the propionylation method was also observed in the propionyl signals and the carbohydrate signals. In other words, the propionyl groups in the TFAA- and DMAP-propionylated wood specimens had longer *T*_1_H values, whereas those in the SA-propionylated wood had shorter *T*_1_H values (Figure 14b). For woody materials, ^1^H spin-lattice relaxation in the laboratory frame (*T*_1_H relaxation) mainly occurs not only via water molecules but also via the portion having the shortest *T*_1_H value, such as lignin [25]. For the propionylated woods manufactured in the present study, however, the *T*_1_H relaxation mainly occurred via the CH_3_ proton in the propionyl group, as indicated by the same tendency between the carbohydrates and propionyl group. For the carbohydrate signals, the *T*_1_H value of the DMAP-propionylated wood increased with increasing propionyl rate (**DMAP-W_1** and **DMAP-W_2**), although the *T*_1_H value of the SA-propionylated wood decreased slightly with increasing propionyl rate (**SA-W_3** and **SA-W_4**). For the propionyl signals, however, the *T*_1_H values of both the DMAP- and SA-propionylated wood specimens increased with increasing propionylation rate. As shown in Appendix A, the *T*_1_H value of SA-propionylated wood increased at a low propionylation rate (**SA-W_1**), and then the *T*_1_H value of carbohydrates gradually decreased with increasing propionylation rate, whereas the *T*_1_H value of the propionyl group rapidly decreased with a slight increase in the propionylation rate (**SA-W_2**).

Combining the above *T*_1_H results and the solid-state NMR spectral analysis, the change in the biomass constituent polymers occurred with each propionylation method as follows. The TFAA treatment broke down most of the cellulose portion and a considerable amount of the lignin portion in the wood; as a result, the propionyl group was mainly introduced into the hemicellulose portion. The introduction of the propionyl group into hemicellulose resulted in the hydrophobicity observed in the ^1^H MAS NMR, and the removal of the lignin portion prolonged the *T*_1_H value. The removal of the cellulose and lignin portions also resulted in swelling in the T-direction and contraction in the L-direction despite the increasing weight. Next, the DMAP treatment reacted more mildly with the constituent polymers in wood, as indicated by the fewer changes in the solid-state NMR spectra of the DMAP-propionylated wood. According to the lower hydrophobicity in the ^1^H MAS NMR, the propionyl group was mainly introduced into the hemicellulose portion, similar to the TFAA-propionylated wood. For the DMAP-propionylated wood, the *T*_1_H value increased with increasing propionylation rate. As the hierarchical structure is composed of constituent polymers in wood, it is disturbed by chemical modifications such as propionylation. This disturbed hierarchical structure weakened the interactions between the constituent polymers, resulting in an increase in the *T*_1_H value. An increase in *T*_1_H due to chemical modification has also been observed not only in acetylation but also in resin-impregnated cypresses [26]. In addition, for the TFAA-propionylated wood, the *T*_1_H increase could be caused not only by delignification but also by the disturbed hierarchical structure.

On the other hand, the propionylation of wood with SA proceeded in multiple stages according to the integrated analysis of the solid-state NMR spectra and the relaxation time. In the first stage, propionylation proceeded mildly, resulting in an increase in the *T*_1_H value, similar to the propionylation with DMAP (**SA-W_1**). Next, decomposition of carbohydrates occurred at the same time as propionylation (**SA-W_2** and **SA-W_3**). In this stage, propionylation mainly occurred in the lignin portion, resulting in lower hydrophobicity as well as a decrease in the *T*_1_H and molecular mobility. Finally, most of the cellulose was removed and the propionylation of hemicellulose increased the hydrophobicity and fluidity of the wood. Because the treatment of wood with SA removed not only the cellulose portion but also the hemicellulose portion, the increase rate was lower even at the highest propionylation rate and the contraction was larger with the occurrence of fine cracks in the SA treatment.

As shown above, the thermoplasticity was significantly influenced by the degradation of the cellulose portion, even though the composition of the biomass constituents in wood and their hierarchical structure depended on the propionylation method. These changes in the composition and hierarchical structure also affected the hydrophobicity and dimensional changes. Understanding the functional expression mechanism will allow for the production of wood-based materials with new characteristics. In the future, we plan to manufacture novel woody materials using chemical modifications based on the expression mechanism of the characteristics.

## 3. Materials and Methods

### 3.1. Original Materials

Pieces of Japanese cypress (*Chamaecyparis obtuse*) (Yamani, Co., Ltd., Nagano, Japan) with dimensions of 5.0 mm (longitudinal direction, L) × 20 mm (radial direction, R) × 20 mm (tangential direction, T) were used as wood specimens. The specimens were cut from a larger piece with a transverse section of 20 mm (R) × 20 mm (T). The initial mass of these specimens was ~0.7 g. Methanol, propionic anhydride (PA), trifluoroacetic anhydride (TFAA), propionic acid, 4-dimethylaminopyridine (DMAP), *N*-methyl pyroridone, sodium hydrogen sulfate, and sulfuric acid (SA) were purchased from FUJIFILM Wako Pure Chemical Corp. (Osaka, Japan) and utilized as obtained.

### 3.2. Preparation of Wood Samples

#### 3.2.1. Degreasing Treatment

To degrease the wood powder, Soxhlet extraction was conducted using methanol for 24 h and then hot water for 24 h. Then, the sample was washed with distilled water and the obtained degreased wood was dried at 35 °C under vacuum for 24 h. This wood specimen was named **DGR-W** and used as a reference for the following propionylation treatments.

#### 3.2.2. Propionylation with Propionic Anhydride

Propionylation with PA was conducted based on the report by Li et al. [27]. Three pieces of degreased wood were placed in a two-neck separable flask (300 mL), and the flask was vacuumed to 400 Pa abs. PA (100 mL) was poured into the flask (vacuum injection), and the flask was then returned to atmospheric pressure. The flask was sealed and reacted at 100 °C for 24 h with gentle stirring. Thereafter, the wood chips were collected, rinsed with 20 mL of methanol, and immersed in 100 mL of methanol to stop any further reaction. After this, Soxhlet extraction was conducted using methanol for 24 h. The wood samples were collected and immersed in 300 mL of pure water. The water was replaced with fresh water every 12 h, and the operation was repeated three times to effectively wash the wood samples. Finally, the propionylated wood samples, called **PA-W** in this study, were dried at 35 °C under vacuum for 24 h.

#### 3.2.3. Propionylation with Trifluoroacetic Anhydride and Propionic Acid

This method was performed based on our previous report [21]. A mixed solution of 50 mL TFAA and 53 mL propionic acid (molar ratio 1:2) was gently stirred for 30 min at 25 °C and used as the propionylation reagent. The procedure for vacuum injection was the same as for **PA-W**. The reaction was carried out at 60 °C for 4 h. The washing and drying procedures were the same as for **PA-W**. The obtained propionylated wood samples are called **TFAA-W** in this study.

#### 3.2.4. Propionylation with Propionic Anhydride and 4-Dimethylaminopyridine

This propionylation method was conducted based on the report by Tanaka et al. [28] with partial modification. A mixed solution of 19.4 mL PA, 90 mL *N*-methyl pyroridone, and 1.8 g DMAP was gently stirred for 30 min at 25 °C and used as the propionylation reagent. The procedure for vacuum injection was the same as for **PA-W**. The reaction was carried out at 40 or 100 °C for 24 h. The washing and drying procedures were the same as for **PA-W**. The obtained samples treated at 40 or 100 °C are called **DMAP-W_1** and **DMAP-W_2**, respectively.

#### 3.2.5. Propionylation with Propionic Anhydride and Sulfuric Acid

Propionylation with PA and SA was conducted based on the report by Shaikh et al. [29]. A mixed solution of 95 mL propionic acid, 38.3 mL PA, 185 mg sodium hydrogen sulfate, and 104 μL SA was gently stirred for 30 min at 25 °C and used as the propionylation reagent. The procedure for vacuum injection was the same as for **PA-W**. The reaction was carried out at 20 °C for 24, 48, and 72 h, or at 40 °C for 24 h. The washing and drying procedures were the same as for **PA-W**. The obtained samples treated at 20 °C for 24, 48, and 72 h are called **SA-W_1**, **SA-W_2**, and **SA-W_3**, respectively. The sample treated at 40 °C is called **SA-W_4**.

### 3.3. Fourier-Transform Infrared (FT-IR) Spectroscopy

FT-IR measurements were performed on a Nicolet 6700 spectrometer (Thermo Scientific Inc., Waltham, MA, USA) with a resolution of 4 cm^−1^ in the standard attenuated total reflectance mode; 32 scans were made in the range of 4000–500 cm^−1^. The measurements were carried out at three or more points on the test piece, and no noticeable differences in the results were observed.

### 3.4. NMR Spectroscopy

Magic-angle spinning (MAS) NMR spectra were acquired on a Varian 400 NMR system spectrometer (Varian Inc., Palo Alto, CA, USA) with a Varian 4 mm double-resonance T3 solid probe. The samples were placed in a 4 mm ZrO_2_ rotor spun at 15 kHz within a temperature range of 20–22 °C. ^1^H MAS NMR spectra were acquired at 399.86 MHz for the ^1^H nuclei and were collected with a 40 ms acquisition period over a 30.5 kHz spectral width in 16 transients, with a 3 s recycle delay. ^13^C MAS NMR spectra were collected with a 2.6 μs π/2 pulse at 100.56 MHz for the ^13^C nuclei and a 40 ms acquisition period over a 30.7 kHz spectral width. Proton decoupling was performed with an 86 kHz ^1^H decoupling radio frequency with a small-phase incremental alteration (SPINAL) decoupling pulse sequence. Cross-polarization and MAS (CP-MAS) NMR studies were conducted with a 5.0 s recycle delay in 1024 transients using a ramped-amplitude pulse sequence with a 2 ms contact time and a 2.6 μs π/2 pulse for the ^1^H nuclei. The amplitude of the ^1^H nuclei was ramped down linearly from 92.6% of its final value during the CP contact time. Pulse saturation transfer and MAS (PST-MAS) NMR measurements were conducted in 2048 transients with a 4.8 μs π/2 pulse for the ^13^C nuclei with a 5 s recycle delay after the saturation of the ^1^H nuclei with 13 consecutive 2.5 μs pulses and a 27.5 μs delay. The ^1^H spin-lattice relaxation time in the laboratory frame (*T*_1_H) was indirectly measured via detection of the ^13^C resonance enhanced by cross-polarization, applied after a π pulse to ^1^H nuclei with the inversion recovery method. 

### 3.5. Hot Compression Test 

A small test piece (5 × 5 × 5 mm) cut from the wood sample was subjected to a compression test (Autograph AG-X plus 20 kN; Shimazu Corp., Kyoto, Japan; load cell SFL-20KNAG; Shimazu Corp., Kyoto, Japan). The sample was placed on a press surface that was heated to 160 °C and pressed in the R-direction at a speed of 1 mm/min. The press was stopped when the force on the specimen reached 5 kN. The apparent compressive stress was calculated based on the initial surface area of the specimen. The areas of the compressed wood specimens were measured using ImageJ image analysis software (U.S. National Institutes of Health, ver. 1.8.0_112, Bethesda, MD, USA) based on the scan images of the compressed specimen. Optical microscopic observation was conducted on a VHX-970F (KEYENCE Corp., Osaka, Japan).

## 4. Conclusions

In this study, the effects of the propionylation method and degree of propionylation on the deformability of wood during heat pressing were investigated. Regardless of the propionylation method, the wood samples were rendered deformable in the T-direction. On the other hand, both the propionylation method and degree of propionylation affected the deformability in the L-direction. Only two samples, which showed a relatively high degree of propionylation by TFAA or SA treatment, exhibited deformability in the L-direction. In these cases, flow in the T-direction occurred under relatively low pressure, and then flow in the L-direction occurred under higher pressure. In contrast, when the wood was propionylated by DMAP treatment or when the degree of propionylation was relatively low even with SA treatment, no flow occurred in the L-direction. To discuss the differences, the chemical components and motility of each sample were evaluated using solid-state NMR measurements. As a result, in the TFAA and SA treatments, the decomposition of the main chain of cellulose was found to progress during propionylation, and this was important for imparting deformability in the L-direction to the wood. These results indicate that the selection of the propionylation method is important for controlling the deformability of wood.

## Figures and Tables

**Figure 1 molecules-26-03539-f001:**
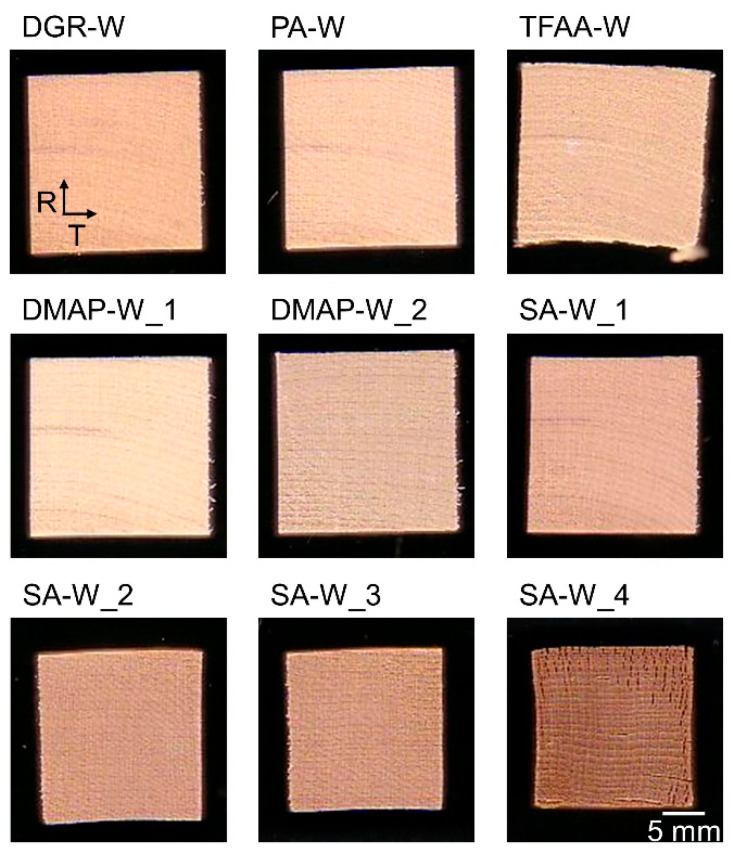
Photographs of degreased and propionylated wood samples.

**Figure 2 molecules-26-03539-f002:**
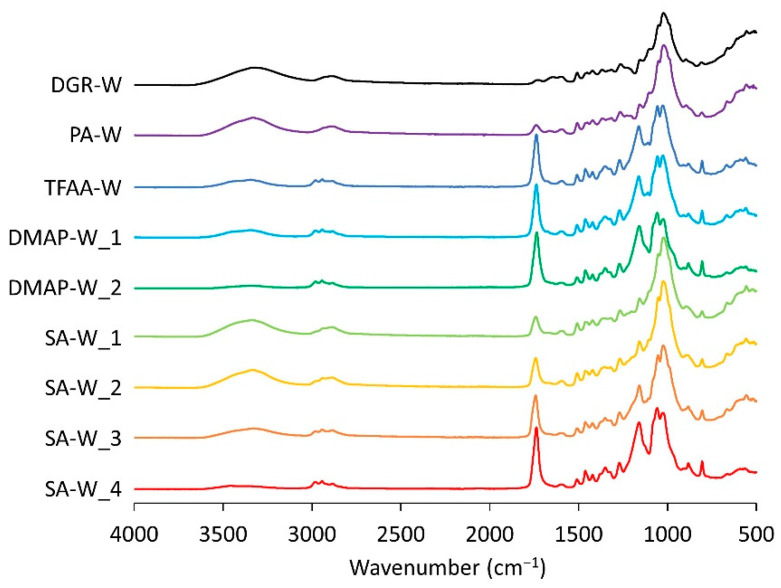
Fourier-transform infrared (FT-IR) spectra of degreased wood and propionylated woods.

**Figure 3 molecules-26-03539-f003:**
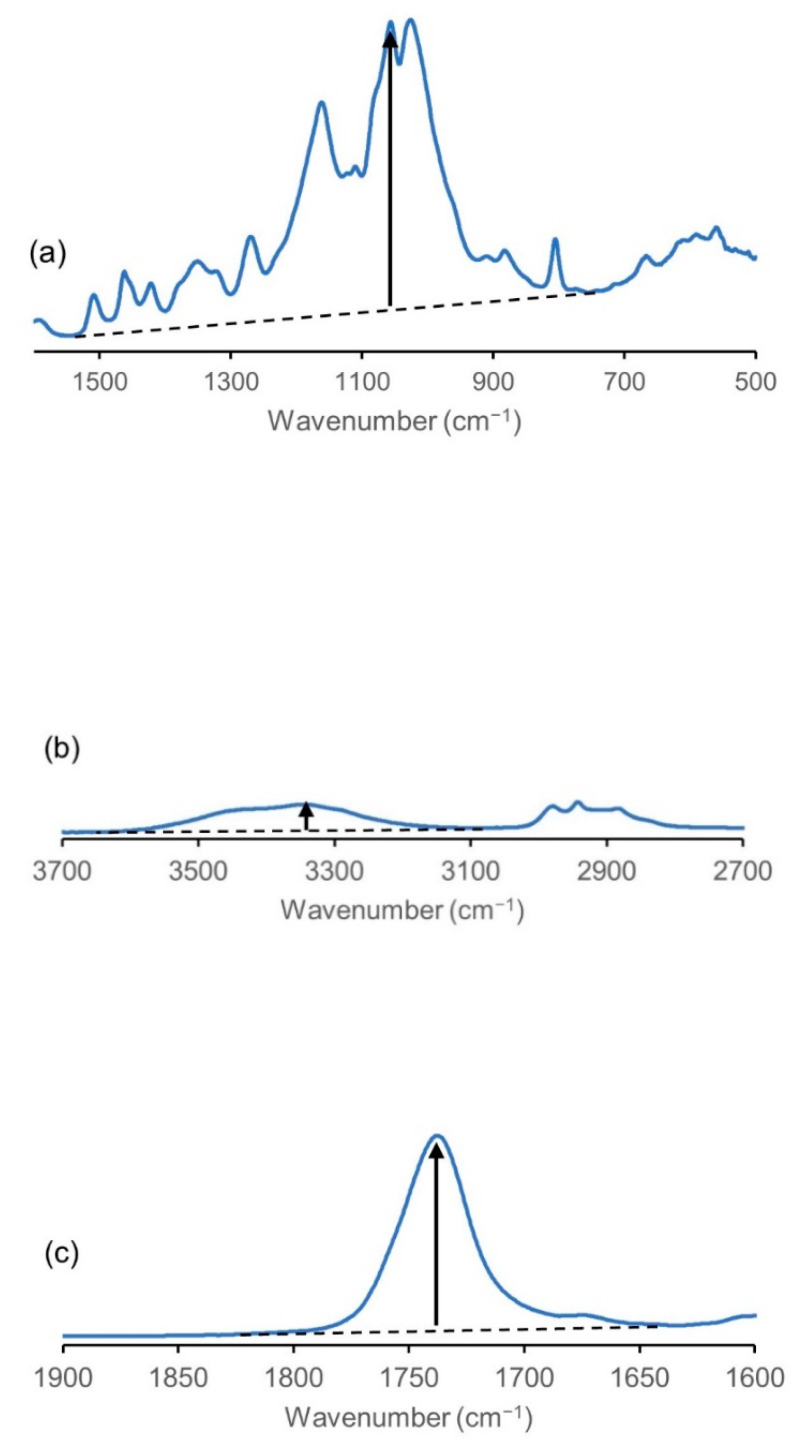
Intensities of the peaks at approximately (**a**) 1060, (**b**) 3340, and (**c**) 1740 cm^−1^ in the IR spectrum of the **TFAA****-W** propionylated wood.

**Figure 4 molecules-26-03539-f004:**
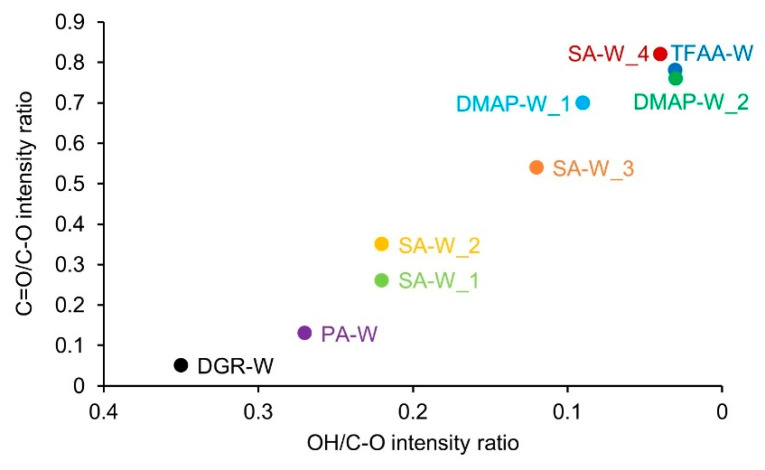
Ratio of the IR peak intensity derived from the C=O groups to that of C–O stretching vibrations (C=O/C–O intensity ratio), showing its dependence on the ratio of the IR peak intensity derived from the OH groups to that of C–O stretching vibrations (OH/C–O intensity ratio).

**Figure 5 molecules-26-03539-f005:**
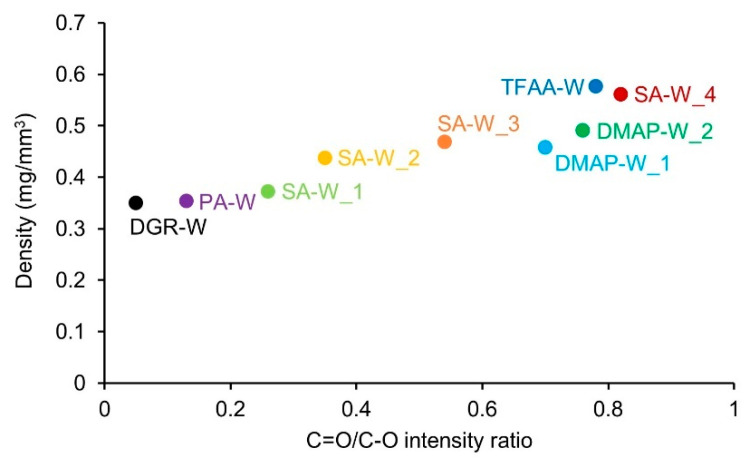
Density of the wood specimens, showing the ratio of the IR peak intensity derived from the C=O groups to that of C–O stretching vibrations.

**Figure 6 molecules-26-03539-f006:**
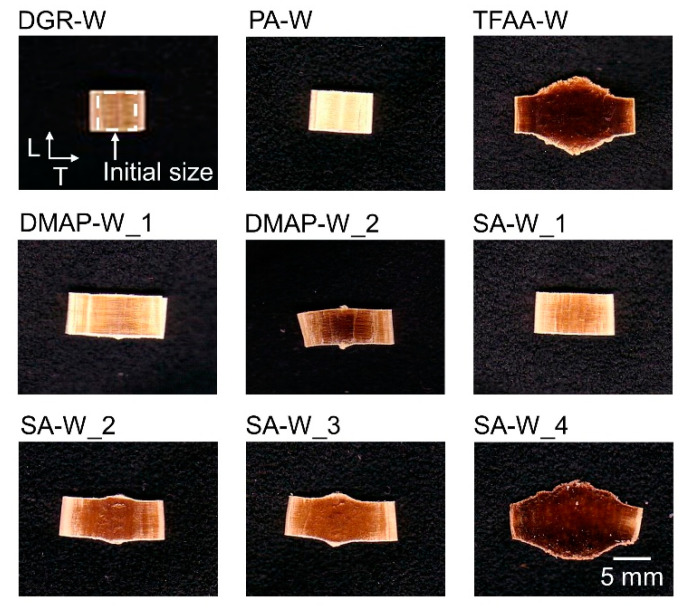
Photographs of degreased and propionylated wood samples after hot compression tests.

**Figure 7 molecules-26-03539-f007:**
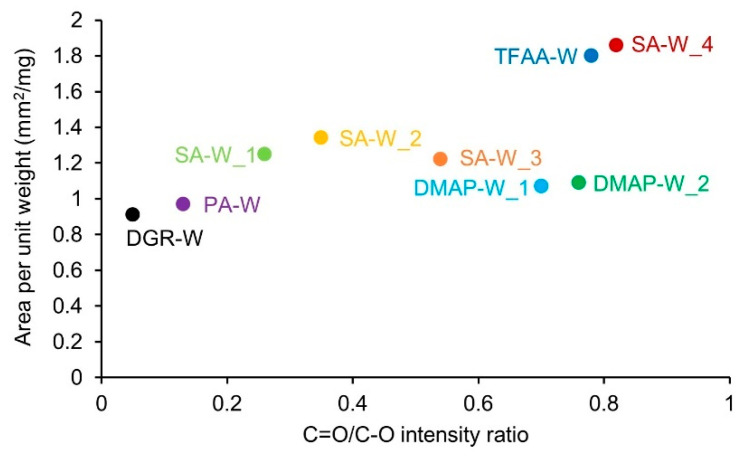
Area per unit weight of the pressed wood specimens, showing the ratio of the IR peak intensity derived from the C=O groups to that of C–O stretching vibrations.

**Figure 8 molecules-26-03539-f008:**
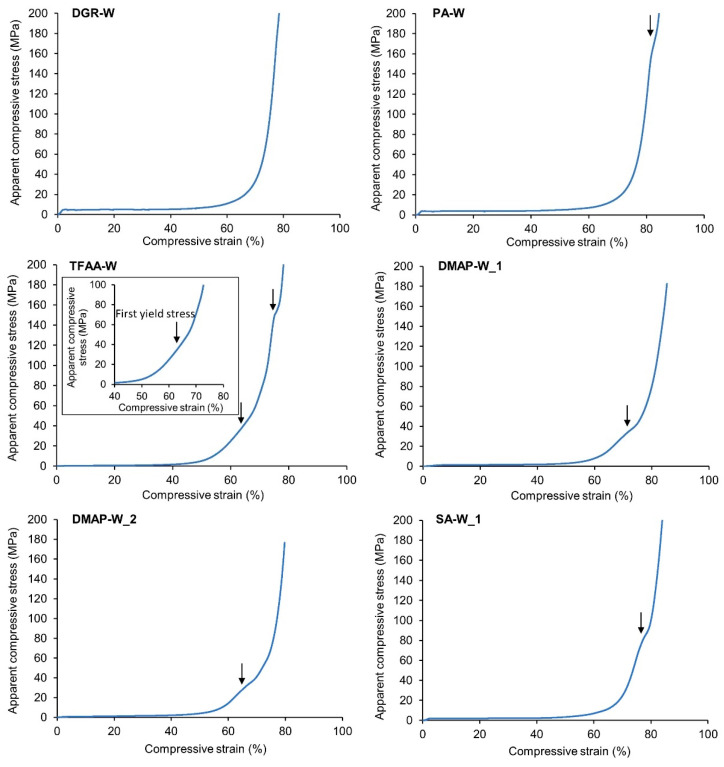
Compressive stress–strain curves of degreased and propionylated wood specimens. The arrows indicate yield stresses. For specimens **TFAA****-W** and **SA****-W_4**, an enlarged view of the stress–strain curve near the first yield stress is also shown.

**Figure 9 molecules-26-03539-f009:**
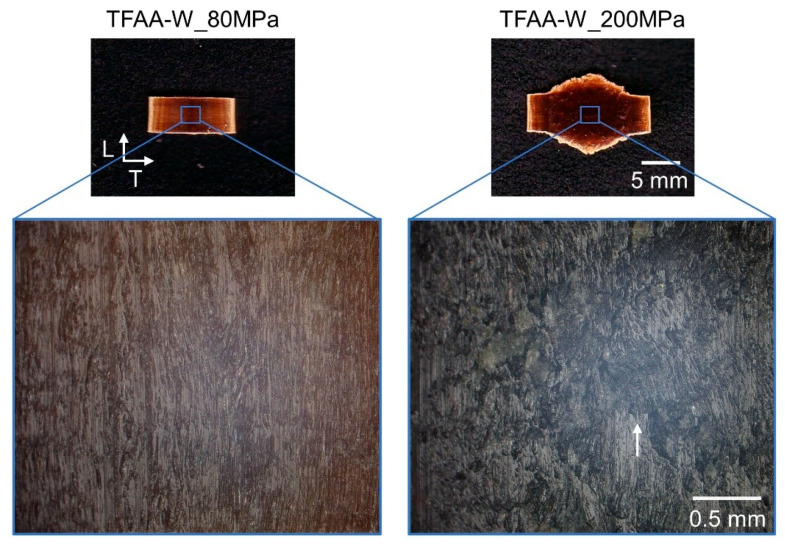
Photographs and microscopic images of **TFAA****-W** specimens after hot compression tests at 80 or 200 MPa. The arrow indicates a typical tear of the wood fibers.

**Figure 10 molecules-26-03539-f010:**
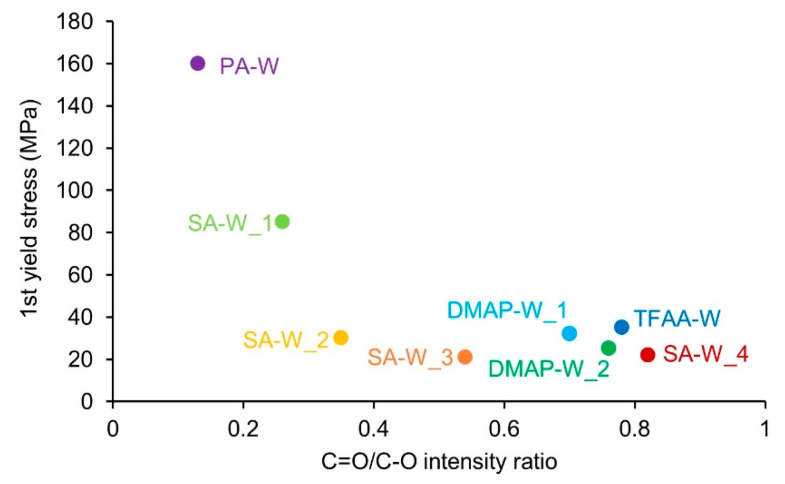
First yield stresses of the propionylated wood specimens, showing the ratio of the IR peak intensity derived from the C=O groups to that of C–O stretching vibrations.

**Figure 11 molecules-26-03539-f011:**
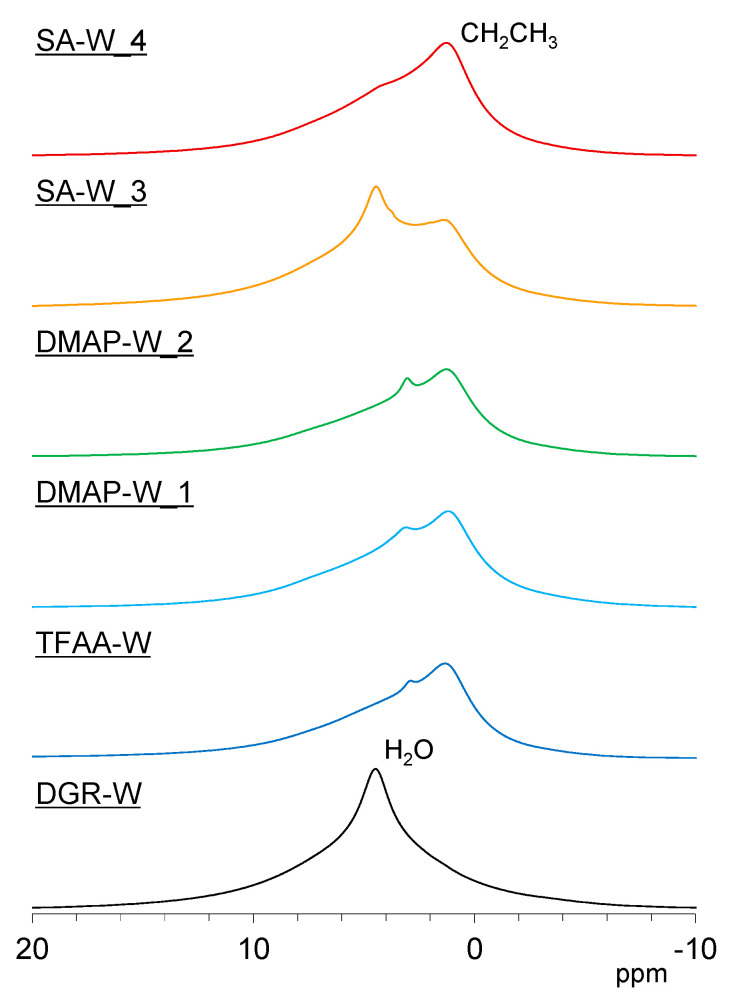
^1^H magic-angle spinning nuclear magnetic resonance (MAS NMR) spectra of degreased wood (black) and propionylated wood samples (**TFAA-W**, blue; **DMAP-W_1**, light blue; **DMAP-W_2**, green; **SA-W_3**, orange; and **SA-W_4**, red).

**Figure 12 molecules-26-03539-f012:**
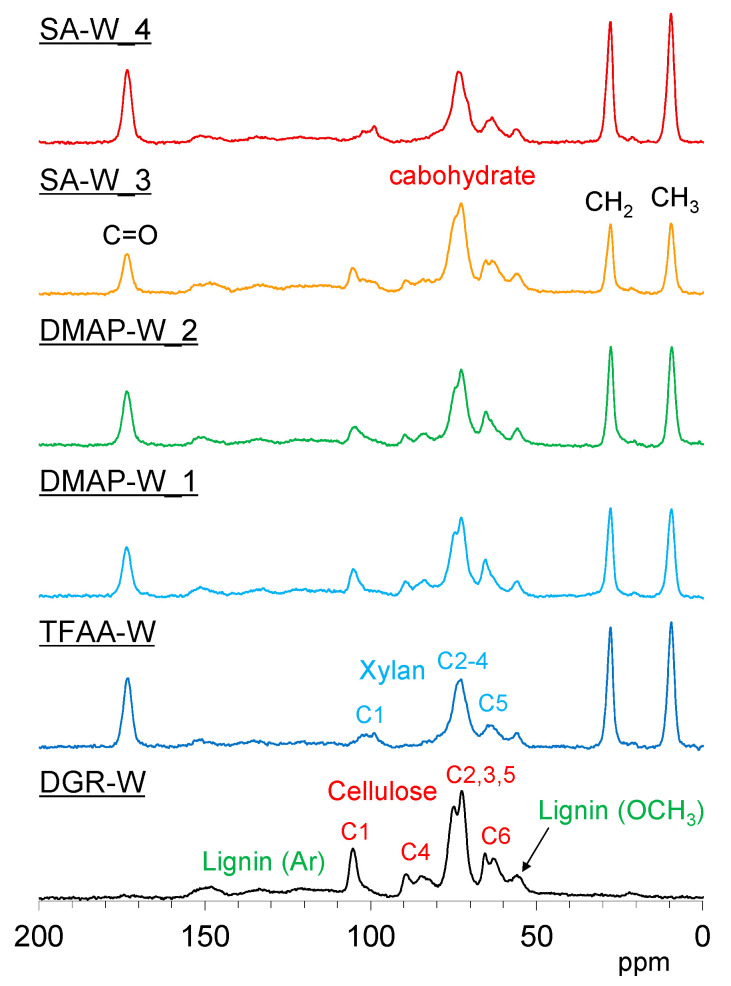
^13^C cross-polymerization (CP)-MAS NMR spectra of degreased wood (black) and propionylated wood samples (**TFAA-W**, blue; **DMAP-W_1**, light blue; **DMAP-W_2**, green; **SA-W_3**, orange; and **SA-W_4**, red).

**Figure 13 molecules-26-03539-f013:**
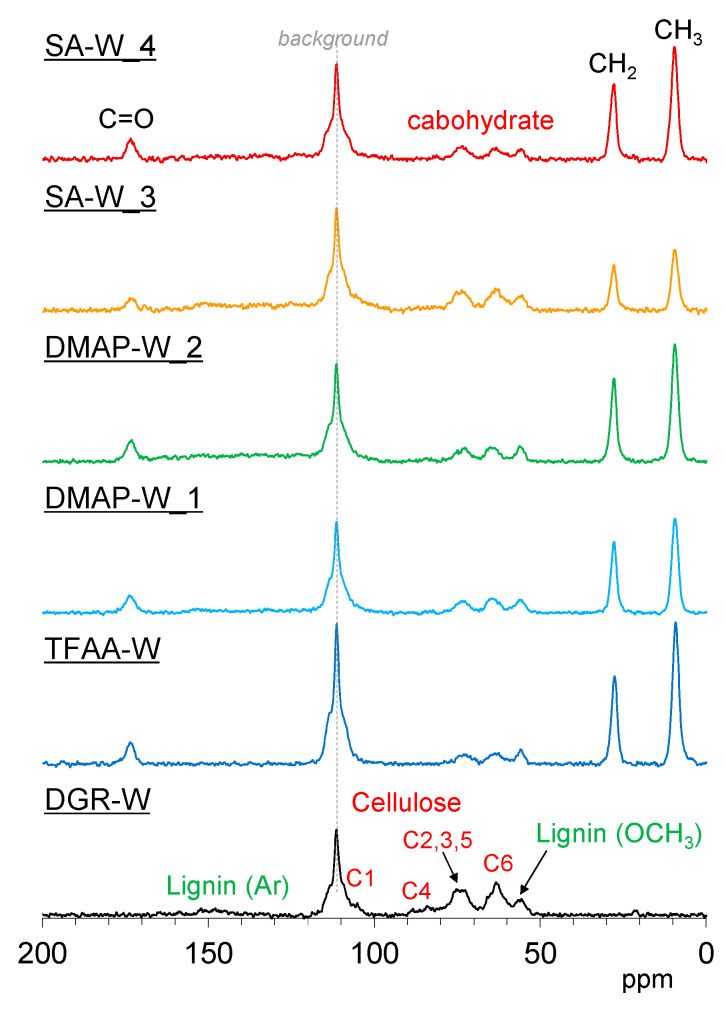
^13^C pulse saturation transfer (PST)-MAS NMR spectra of degreased wood (black) and propionylated wood samples (**TFAA-W**, blue; **DMAP-W_1**, light blue; **DMAP-W_2**, green; **SA-W_3**, orange; and **SA-W_4**, red).

**Figure 14 molecules-26-03539-f014:**
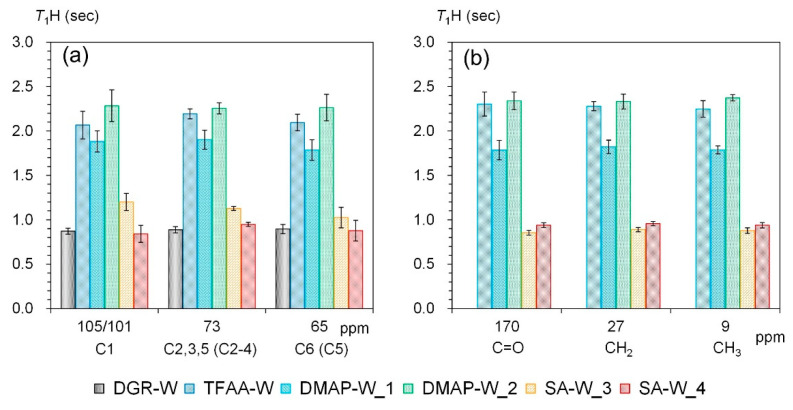
^1^H spin-lattice relaxation time in the laboratory frame (*T*_1_H) of degreased and propionylated wood samples in the dry state: (**a**) carbohydrates, (**b**) propionyl group (error bars show the standard deviation from the mean of five measurements).

**Table 1 molecules-26-03539-t001:** Sample code of the synthesized woods and description of the propionylation methods.

Sample Codes	Propionylation Methods ^a^		
	Regents	Temp. (°C)	Time (h)
**DGR** **-W**	- ^b^	- ^b^	- ^b^
**PA-W**	Propionic anhydride	100	24
**TFAA** **-W**	Trifluoroacetic anhydride/Propionic acid	60	4
**DMAP** **-W** **_1**	4-Dimethylaminopyridine/Propionic anhydride/*N*-methyl pyroridone	40	24
**DMAP** **-W_2**		100	24
**SA** **-W** **_1**	Sulfuric acid/Propionic acid/Propionyl anhydride/Sodium hydrogen sulfate	20	24
**SA** **-W** **_2**		20	48
**SA** **-W** **_3**		20	72
**SA** **-W** **_4**		40	24

^a^ The increase was calculated based on the dry degreased wood pieces before propionylation treatment; ^b^ Without propionylation (Degreased wood).

**Table 2 molecules-26-03539-t002:** Increase in the weight and length of each side of the propionylated wood samples relative to the degreased wood piece.

Samples	Increase (%) ^a^			
	Weight	T ^b^	R ^c^	L ^d^
**PA** **-W**	2.1 ± 0.1	0.1 ± 0.1	0.2 ± 0.0	0.2 ± 0.2
**TFAA** **-W**	58.7 ± 0.2	9.4 ± 0.3	0.8 ± 0.3	−10.4 ± 0.2
**DMAP** **-W** **_1**	40.0 ± 0.1	5.9 ± 0.1	1.5 ± 0.2	0.8 ± 0.2
**DMAP** **-W_2**	45.6 ± 0.1	5.8 ± 0.1	1.5 ± 0.3	0 ± 0.2
**SA** **-W** **_1**	2.4 ± 0.1	−3.1 ± 0.1	−1.9 ± 0.2	0.5 ± 0.1
**SA** **-W** **_2**	10.9 ± 0.1	−5.9 ± 0.2	−5.5 ± 0.3	0.3 ± 0.1
**SA** **-W** **_3**	12.2 ± 0.3	−8.8 ± 0.2	−9.3 ± 0.1	0.5 ± 0.2
**SA** **-W** **_4**	37.9 ± 0.8	−3.7 ± 0.9	−7.2 ± 0.6	−3.4 ± 1.1

^a^ The increase was calculated based on the dry degreased wood pieces before propionylation treatment, and the standard error was calculated based on the values of three samples; ^b^ Tangential direction; ^c^ Radial direction; ^d^ Longitudinal direction.

## Data Availability

The data presented in this study are available upon request from the corresponding author.

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
