# Peer review of "Effect of the Propionylation Method on the Deformability under Thermal Pressure of Block-Shaped Wood"

_molecules, 2021, doi:10.3390/molecules26123539_

Round 1

Reviewer 1 Report

The author produced multiple propio-11 nylated wood specimens by several propionylation methods and elucidated the factors affecting 12 the deformability of the wood. The manuscript was well written and organized.

one question is:

How much percent of water for each specimen was? How did this influnence the mechnical properties?

Author Response

Dear Reviewer,

Thank you for your appreciation.

The author produced multiple propionylated wood specimens by several propionylation methods and elucidated the factors affecting the deformability of the wood. The manuscript was well written and organized.

Ans. Thank you for your positive comments.

one question is:

How much percent of water for each specimen was? How did this influnence the mechnical properties?

Ans. These specimens were dried under vacuum before measurements, so their moisture content was almost 0%. We think that this low moisture content of propionylation samples scarcely affected the hot compression test. We also think that effects of moisture content on mechanical properties are important issues to be examined in the future.

Reviewer 2 Report

  1. The paper entitled ‘Effect of the propionylation method on the deformability under thermal pressure of block-shaped wood’ is focused on deformability of propionylated cellulose. The paper is well organized and written. Obtained by Authors results are well presented and described.

    Below you can find some comments:

    1. Please add to the paper the new table with samples' symbols and description.
    2. Symbols T, R and L presented in Table 1 should be explained.
    3. Did you check the chemical stability of obtained materials in different organic solvents??

Author Response

Dear Reviewer,

Thank you for your appreciation. We revised our manuscript according to your helpful comments.

The paper entitled ‘Effect of the propionylation method on the deformability under thermal pressure of block-shaped wood’ is focused on deformability of propionylated cellulose. The paper is well organized and written. Obtained by Authors results are well presented and described.

Ans. Thank you for your positive comments.

Please add to the paper the new table with samples' symbols and description.

Ans. We added a following table in the Introduction section to explain the samples’ symbols and description.

Table 1. Sample code of the synthesized woods and description of the propionylation methods.

Sample

Codes

Propionylation methodsa

Regents

Temp. (°C)

Time (h)

DGR-W

-b

-b

-b

PA-W

Propionic anhydride

100

24

TFAA-W

Trifluoroacetic anhydride/Propionic acid

60

4

DMAP-W_1

4-Dimethylaminopyridine/Propionic anhydride/N-methyl pyroridone

40

24

DMAP-W_2

100

24

SA-W_1

Sulfuric acid/Propionic acid/Propionyl anhydride/Sodium hydrogen sulfate

20

24

SA-W_2

20

48

SA-W_3

20

72

SA-W_4

40

24

a The increase was calculated based on the dry degreased wood pieces before propionylation treatment.; b Without propionylation (Degreased wood).

Symbols T, R and L presented in Table 1 should be explained.

Ans. Description of these symbols were added to the Table. The revised Table is as follows.

Table 2. Increase in the weight and length of each side of the propionylated wood samples relative to the degreased wood piece.

Samples

Increase (%)a

Weight

Tb

Rc

Ld

PA-W

2.1 ± 0.1

0.1 ± 0.1

0.2 ± 0.0

0.2 ± 0.2

TFAA-W

58.7 ± 0.2

9.4 ± 0.3

0.8 ± 0.3

−10.4 ± 0.2

DMAP-W_1

40.0 ± 0.1

5.9 ± 0.1

1.5 ± 0.2

0.8 ± 0.2

DMAP-W_2

45.6 ± 0.1

5.8 ± 0.1

1.5 ± 0.3

0 ± 0.2

SA-W_1

2.4 ± 0.1

−3.1 ± 0.1

−1.9 ± 0.2

0.5 ± 0.1

SA-W_2

10.9 ± 0.1

−5.9 ± 0.2

−5.5 ± 0.3

0.3 ± 0.1

SA-W_3

12.2 ± 0.3

−8.8 ± 0.2

−9.3 ± 0.1

0.5 ± 0.2

SA-W_4

37.9 ± 0.8

−3.7 ± 0.9

−7.2 ± 0.6

−3.4 ± 1.1

a The increase was calculated based on the dry degreased wood pieces before propionylation treatment. The standard error was calculated based on the values of three samples.; b Tangential direction; c Radial direction; d Longitudinal direction.

Did you check the chemical stability of obtained materials in different organic solvents??

Ans. In this paper, the organic solvent resistance of the propionylated wood have not been investigated. This will be one of the future issues. Thank you for your valuable advice.